# Clinical Applications of Magnetic Resonance-Guided Radiotherapy: A Narrative Review

**DOI:** 10.3390/cancers15112916

**Published:** 2023-05-26

**Authors:** Colton Ladbury, Arya Amini, Amanda Schwer, An Liu, Terence Williams, Percy Lee

**Affiliations:** 1Department of Radiation Oncology, City of Hope National Medical Center, Duarte, CA 91010, USA; cladbury@coh.org (C.L.);; 2Department of Radiation Oncology, City of Hope Orange County Lennar Foundation Cancer Center, Irvine, CA 92618, USA

**Keywords:** magnetic resonance-guided radiotherapy (MRgRT), image-guided radiation therapy (IGRT), magnetic resonance imaging (MRI)

## Abstract

**Simple Summary:**

Magnetic resonance-guided radiotherapy (MRgRT) is an emerging radiotherapy technology combining real-time magnetic resonance imaging and radiation delivery. By administering radiation with a linear accelerator with built in low-field or high-field MRI, practitioners have a greater ability to align to the target for daily set-up, precisely track the motion of and thereby target or avoid tissues and adapt to inter-treatment daily changes. This decreased uncertainty has implications for facilitating smaller, less-toxic treatment margins, potentially allowing delivery of higher dose radiotherapy that will lead to better control of tumors. The technology has already found success in treating breast, prostate, pancreatic, liver, lung, and limited metastatic cancers, in addition to non-oncologic indications such as cardiac ablation. The present narrative review aims to describe the current and future state of MRgRT technology and research.

**Abstract:**

Magnetic resonance-guided radiotherapy (MRgRT) represents a promising new image guidance technology for radiation treatment delivery combining an onboard MRI scanner with radiation delivery technology. By enabling real-time low-field or high-field MRI acquisition, it facilitates improved soft tissue delineation, adaptive treatment, and motion management. Now that MRgRT has been available for nearly a decade, research has shown the technology can be used to effectively shrink treatment margins to either decrease toxicity (in breast, prostate cancer, and pancreatic cancer) or facilitate dose-escalation and improved oncologic outcomes (in pancreatic and liver cancer), as well as enabling indications that require clear soft tissue delineation and gating (lung and cardiac ablation). In doing so, the use of MRgRT has the potential to significantly improve the outcomes and quality of life of the patients it treats. The present narrative review aims to describe the rationale for MRgRT, the current and forthcoming state of technology, existing studies, and future directions for the advancement of MRgRT, including associated challenges.

## 1. Introduction

Over the last twenty years, advances in the delivery of conformal radiotherapy, namely through the widespread adoption of intensity-modulated radiotherapy (IMRT) and stereotactic body radiation therapy (SBRT), have facilitated improvements in both tumor control and toxicity. For such conformal radiotherapy to be feasible, it has necessitated advances in image-guided radiotherapy (IGRT), which can help ensure the radiation target volumes are treated while adjacent organs-at-risk (OARs) are spared.

Magnetic resonance imaging (MRI) is an imaging modality that has proven advantages for radiation treatment by aiding in delineating relevant structures since it offers superior soft tissue contrast. Previously, the incorporation of MRI into IGRT technologies has been prohibited due to the magnetic field impacting the shape of the radiation beam via the electron return effect, as well as electronic components of the linear accelerator deteriorating the quality of the MR imaging [1]. However, recent advances in technology have been able to overcome that challenge and have enabled the development of integrated MR-guided radiotherapy (MRgRT) systems [2].

A number of MRgRT systems have been developed, including the MRIdian by ViewRay and Unity by Elekta, incorporating both low-field and high-field MRI imaging. These systems are specifically designed to account for the impact of the magnetic field on the trajectory of charged particles (via the Lorenz force) in their dose calculation algorithms [3]. MRgRT systems offer several advantages compared to other IGRT options including improved soft tissue visualization (which improves target delineation and motion management while enabling dose escalation and adaptive treatments), no radiation exposure (which enables more frequent imaging), real-time monitoring (obviating the need for fiducials), and ability to assess treatment response over the course of treatment. Consequently, tumors with significant motion, tumors with better delineation on MRI, tumors whose treatment benefits from smaller margins, and tumors that benefit from hypofractionation all stand to benefit from MRgRT. The overall clinical workflow of MRgRT is summarized in Figure 1. This review aims to provide a brief overview of MRgRT technology and associated clinical indications, with supporting evidence.

## 2. Types of MRgRT

### 2.1. Low-Field

Low-field MRI-guided radiotherapy uses a magnetic field strength of 0.2 to 1.0 Tesla [4]. Due to the lower magnetic field strength of low-field, the image resolution is typically lower compared to high-field. A primary example of a low-field MR-linac is the ViewRay MRIdian. Key points regarding low-field MRgRT are summarized in Table 1.

### 2.2. High Field

High-field MRI-guided radiotherapy uses a magnetic field strength of greater than 1.0 Tesla. A primary example of a high-field MR-linac is the Elekta Unity. Key points regarding high-field MRgRT are summarized in Table 1. Due to the higher image quality, if available, the use of a high-field rather than a low-field MR-linac will generally be preferable, as it will better facilitate the accurate delineation of structures of interest.

### 2.3. MagnetTx Aurora RT

Distinct from current MR-linac low-field or high-field options is the MagnetTx Aurora RT system that has recently been approved by the Food and Drug Administration. This system consists of a larger 110 × 60 cm bore and an 0.5 T open biplanar magnet [5]. The configuration allows for configurations where the radiation is either perpendicular or parallel to the MR-linac’s main magnetic field. The parallel configuration is distinct from other systems and limits exit skin dose and dosimetric hotspots associated with the electron return effect and electron streaming effect [6,7]. The geometry also creates a more homogeneous dose distribution [8]. At present, the MagnetTx system has not been used nearly as much as the MRIdian or Unity systems, so future work as more units are installed will further elucidate the clinical benefit of the system.

## 3. Indications Treated by MRgRT

The way in which MRgRT can improve radiotherapy is dependent on the disease site; Table 2 summarizes, by disease site, rationales for how MRgRT can benefit clinical management. These disease sites and their clinical evidence are summarized below. The indications summarized below by no means represent a comprehensive list, as MRgRT has potential for most disease sites (including head and neck [9], colorectal [10], brain [11], and gynecological cancers [12,13,14]) due to similar rationales to the included indications.

### 3.1. Prostate

The primary rationale for the use of MRgRT in prostate cancer has been in the mitigation of toxicity. Based on the available literature, this is accomplished by aiming to spare either the bladder, urethra, small and large bowel, or rectum through reduced PTV margins or through avoidance of the urethra by delineating it as an avoidance structure within the prostate. The use of an integrated MRgRT system is preferable to workflows wherein a planning MRI is fused to a planning CT because it reduces uncertainties associated with the MRI-CT fusion, does not require fiducial marker placement, and allows physicians to take into account daily changes in the prostate, seminal vesicle, and rectal anatomy [17].

The reduction of PTV margins in prostate cancer using MRgRT is the indication with the highest level of evidence based on the MIRAGE randomized phase III trial [15]. As part of the MIRAGE trial, 154 patients with localized prostate cancer were randomized to receive either standard CT-guided SBRT (CTgRT) with a 4 mm PTV margin or MRgRT with a 2 mm PTV margin. In both arms, the prostate PTV received 40 Gy in 5 fractions, with escalation to 42 Gy allowing for dominant intraprostatic nodules. Elective nodal regions received 25 Gy while gross nodes received 35 Gy. Hydrogel placement was permitted at the investigator’s discretion but was not required. The study met its primary objective, with the incidence of physician-reported grade ≥ 2 GU toxicity being 43.4% in the CTgRT arm versus 24.4% in the MRgRT arm (*p* = 0.006). This was primarily driven by reductions in urinary frequency and urinary retention. There was also a reduction in grade ≥ 2 GI toxicity from 10.5% in the CTgRT to 0% in the MRgRT arm (*p* = 0.001), primarily driven by reductions in diarrhea and proctitis. Patient-reported outcomes measured by the International Prostate Symptom Score (IPSS) and Expanded Prostate Cancer Index (EPIC) bowel scores were also consistent with these findings. It is important to note that a difference in toxicity between the two arms is not overly surprising due to the smaller margins in the MRgRT arm. The main takehome from the trial is that MRgRT enabled aggressive margin reduction, due to increased confidence in targeting, which would be more challenging using only CT.

The use of MRgRT to spare the urethra was evaluated as part of a phase II study of 101 patients by Bruynzeel et al. [16]. In this study, patients with intact prostates were treated with SBRT to a total dose of 36.25 Gy with daily treatment plan adaptation and sparing of the urethra to 32.5 Gy. As with the MIRAGE trial, smaller PTV margins (3 mm) were used. With CT-guided SBRT, there is little confidence in the daily position of the urethra, and therefore it is challenging to spare it under those workflows. But using MRgRT and daily adaptation it becomes possible, optimally leading to decreased GU toxicity. The rates of grade ≥ 2 GU or GI toxicity were 23.8% and 5.0%, respectively. No early grade ≥ 3 GU or GI toxicity was observed. These rates are overall similar to those seen in the MIRAGE trial, despite slightly larger margins and no rectal spacers, and therefore urethral sparing, in combination with the MIRAGE technique, may prove to be a valuable means of reducing toxicity in patients with intact prostate cancer.

Unsurprisingly, as the role of SBRT for salvage radiotherapy for prostate cancer is not standard, the literature on the use of MRgRT for that indication is more limited. However, it was used in approximately a third of patients treated in the non-randomized phase II SCIMITAR trial [35]. In this trial, 69 patients received CTgRT and 31 received MRgRT. Patients receiving CTgRT had 5 mm PTV margins on the prostate bed while patients treated with MRgRT had 3 mm PTV margins. Elective nodal and gross nodal boost PTV margins were 5 mm and 3 mm, respectively, regardless of platform. Dose levels for both platforms were 30–34 Gy to the prostate bed, 25 Gy to elective nodes, and 35 to 40 Gy to gross nodes, all delivered in 5 fractions. Online adaptive radiotherapy was only administered for 4 total fractions. MRgRT was associated with significantly lower acute GI toxicity (*p* = 0.006), with 8.6% of patients in the CTgRT arm experiencing grade ≥ 2 GI toxicity compared to 0% in the MRgRT arm. Again, this is not overly surprising due to the smaller margins, and it is not possible to make strong conclusions in the absence of randomization. Nevertheless, these data are compelling and form the basis for the phase II EXCALIBUR study (NCT04915508), which will only include MRgRT.

### 3.2. Pancreas

In pancreatic cancer, MRgRT has the potential to help overcome challenges that have limited the role of radiotherapy in its management. Pancreatic cancer tends to be a radioresistant histology and therefore stands to benefit from high doses. However, due to proximity to nearby OARs such as the duodenum, stomach, and small bowel using CT-based planning and delivery, conformality has been challenging due to insufficient soft tissue contrast to differentiate between tumor and adjacent organs at risk on the on-board cone beam CT, which limits the sparing of healthy tissues. Lastly, due to the proximity to the diaphragm, motion management techniques are helpful. All of those limitations can be addressed using MRgRT, and in doing so can facilitate dose escalation.

The role of MRgRT in pancreatic cancer was investigated as part of a multi-institutional study by Rudra et al. [18]. In this study of 44 patients with inoperable pancreatic cancer, 13 patients received conventional fractionation (40–55 Gy in 25–28 fractions), 9 received hypofractionation (50–67.5 Gy in 10–15 fractions), 6 received conventional SBRT (30–35 Gy in 5 fractions), and 16 received high-dose SBRT (40–52 Gy in 5 fractions). Adaptive replanning was used for treatment courses of 15 or fewer fractions, with OAR doses prioritized and dose escalation permitted if normal tissue constraints could be met. For analyses, patients were stratified by the prescription biologically effective dose using an alpha/beta ratio of 10 (BED_10_) of >70 Gy, which was termed the “high-dose cohort” and the “standard-dose” cohort, which was ≤70 Gy. Two-year overall survival (OS) in the high-dose vs standard-dose groups was 49% vs. 30%, respectively (*p* = 0.03). Two-year local control (LC) was 77% for the high-dose group vs. 57% (*p* = 0.15). Importantly, grade ≥ 3 GI occurred only in three patients, all of whom were in the standard-dose cohort and received concurrent gemcitabine. It was hypothesized that the reason for reduced GI toxicity in the high-dose cohort was due to a higher percentage of patients in the high-dose cohort requiring on-table adaptive replanning to meet the OAR dose constraints for the viscous hollow organs.

Based on that data, the phase II stereotactic MRI-guided adaptive radiation therapy (SMART) trial was designed [19]. The trial treated 136 patients with borderline resectable or unresectable locally advanced pancreatic cancer to a total dose of 50 Gy in 5 fractions following a minimum of 3 months of chemotherapy. The trial was powered to detect an approximately 8% reduction of grade ≥ 3 GI toxicity, from 15.8% in a historical control to 8% at 90 days. The incidence of acute grade ≥ 3 GI toxicity at least definitely, probably, and possibly related to SMART was 0%, 2.2% (n = 3) and 8.8% (n = 12), respectively. One-year OS, LC and distant progression-free survival (DPFS) from SMART were 93.9%, 82.9% and 50.6%, respectively. Thus, with SMART it appears feasible to deliver ablative doses of radiotherapy for locally advanced pancreatic cancer without excessive acute GI toxicity. Longer follow-up will be required to assess for late toxicity and long-term oncological outcomes. For comparative efficacy relative to the current standard of care (chemotherapy alone), the LAP-ABLATE trial is currently enrolling (NCT05585554).

### 3.3. Liver

The rationale for the use of MRgRT for liver cancer is similar to pancreatic cancer: tumors can be difficult to visualize on CT imaging, there is significant motion associated with the respiratory cycle, and tumors located in the central and peripheral liver are close to OARs. Related to those factors, fiducials are routinely used for CT-based treatment, which is an invasive procedure that carries a risk of infection and bleeding. Further, these fiducials can cause metal artifacts, which can further limit target delineation. All of those challenges are suitably addressed with MRgRT. Under MRgRT workflows, the use of liver-directed contrast (Eovist) is an additional option that can further enhance the visualization of the liver tumor at simulation and daily treatments for on-table adaptive replanning and tumor tracking for motion management [10,36].

In a multi-institutional study by Rosenberg et al., 26 patients with hepatic tumors were treated with MRgRT [20]. Of these patients, six had hepatocellular carcinoma, two had cholangiocarcinoma, and 18 had metastatic lesions (with colorectal cancer being the predominant source). Radiation treatment volumes included the gross tumor volume and a 2 to 5 mm PTV expansion at the treating physician’s discretion. The median dose was 50 Gy in 5 fractions, though doses as low as 30 Gy with fractional doses as low as 6 Gy were permitted to be included in the analysis. Adaptive replanning was not used. Two-year LC and OS were 80.4% and 60%, respectively. LC was 100% for patients with hepatocellular carcinoma histology. Grade ≥ 3 GI toxicity was observed in 7.7% of patients, attributed to a hilar stricture and portal hypertension. There is no grade 4 or 5 toxicity observed.

A separate single-institutional series of 17 patients with locally advanced cholangiocarcinoma was performed by Luterstein et al. [21]. Twelve patients had extrahepatic cholangiocarcinoma and five patients had intrahepatic tumors. Again, radiation treatment volumes included the gross tumor volume and a 3 mm PTV expansion. The radiation dose was 40 Gray in 5 fractions in the majority of patients and patients were treated with a breath-hold respiratory gated technique. In this study, adaptive planning was routinely implemented with the protection of surrounding radiosensitive OARs superseding PTV coverage. Reoptimization occurred if this new geometry predicted a normal tissue overdose. Two-year LC and OS were 73.3% and 46.1%, respectively. One patient experienced a grade 3 duodenal ulcer and one patient experienced a late grade 2 gastritis/colitis, but there was otherwise no grade > 1 toxicity.

Based on these promising series, a phase I trial was performed by Van Dams et al. [22]. In this study of 20 patients (8 with primary liver tumors and 12 with metastatic liver tumors) and 25 total tumors, patients were treated with a 3 mm margin on gross tumor volume. Fractionation regimens included 45 Gray in 3 fractions, 48 Gray in 3 fractions, and 54 Gray in 3 fractions for single lesion plans and 40 Gray in 5 fractions, 50 Gray in 5 fractions, and 60 Gray in 5 fractions for multi-lesion plans. Adaptive replanning was not performed in the study. Two-year LC and OS were 79.6% and 50.7%, respectively. When stratifying by a BED of ≥100 Gy, LC was 93.8% versus 33.3%. Similarly, LC was 93.3% in patients with a single lesion compared to 33.3% in patients with multiple lesions. No acute grade ≥ 3 toxicity was observed but there was 1 single case of a late grade 3 duodenal ulceration that progressed to grade 4 sepsis, which was felt to be due to a highV35_Gy_ from an adjacent loop of small bowel that received V35_Gy_ of 0.46 cm^3^. Subsequently, a maximum constraint of V35_Gy_ less than 0.35 cm^3^ was implemented and no further late grade ≥ 3 toxicity was observed. Based on these data, it appears that MRgRT is able to deliver ablative doses for liver tumors with high rates of local control. A randomized three-arm phase 2 study of 80 patients with hepatic metastases, MAESTRO, is ongoing and is aiming to assess whether delivery of a BED ≥ 100 Gy is feasible using either MRgRT or an internal target volume (ITV)-based standard SBRT. The third arm will be for patients whose anatomy makes it impossible to achieve target dose constraints with a BED of ≥100 Gy and will be treated with MRgRT at the highest possible dose.

### 3.4. Breast

The role of MRgRT in the management of breast cancer is emerging, as there is a growing amount of data supporting partial breast rather than whole breast irradiation. Accelerated partial breast irradiation (APBI) is increasingly being implemented due to favorable toxicity profiles and convenience. However, since the whole breast is no longer being treated, delivering effective APBI requires accurate delineation of the postoperative breast cavity. Due to subtle differences in soft tissue densities, this can be challenging on CT, and therefore often requires margins of at least a centimeter, which can reduce the cosmetic benefit. MRgRT can better visualize the lumpectomy cavity, which can better enable smaller margins. Further, as with other disease sites, MRgRT can help address motion management associated with the breathing cycle as well as daily setup areas associated with breast positioning.

MRgRT for APBI was evaluated by Kennedy et al. as part of the phase I/II trial [25]. As part of this trial, they evaluated the feasibility of delivering APBI and a single postoperative fraction to 50 women with low-risk, hormone-positive breast cancer. Notably, this is distinct from most other APBI regimens, which typically are a minimum of 5 fractions. Thus, this regimen most closely resembles a regimen of intraoperative radiotherapy. All but two patients were treated with MRgRT. The primary treatment volume entailed the surgical bed with no expansion, limited to 5 mm from the skin. An additional treatment volume was derived from the surgical bed plus a 1 cm margin. The surgical bed was planned to a total dose of 20 Gy while the surgical bed plus 1 cm was planned to a minimum dose of 5 Gy. At a median follow-up of 25 months, there was only one noninvasive ipsilateral breast tumor recurrence event. No acute grade ≥ 3 toxicity was observed and there was only a single case of grade 2 chest wall pain. There was also no adverse impact on quality of life or cosmesis.

Another variation of an APBI was assessed in a single-arm prospective trial by Vasmel et al. [26]. In this study, 36 patients with unifocal, nonlobular, node-negative tumors measured a maximum of 20 mm in patients between 50 and 70 years old or 30 mm in patients older than 70 years old. In this case, the single ablative dose was given prior to surgery, which is another interesting departure from a standard of care, as radiation is typically given following surgery. The gross tumor volume was treated to a total dose of 20 Gy, while the breast tumor volume plus a 2 cm margin was treated to a minimum dose of 15 Gy. A pathologic complete response was observed in 15 (42%) patients, while a radiologic complete response was observed in 15 patients, 10 of whom had a pathologic complete response, corresponding to a positive predictive value of 67%. At a median follow-up of 21 months, all patients developed grade 1 fibrosis, and 31% and 3% of patients, respectively, experienced grade 2 and 3 toxicity. The grade 2 and 3 toxicity were driven in large part by 14% and 3% of patients, respectively, who developed wound infection, which one could hypothesize because of the sequencing of radiation and surgery. There were no local recurrences or deterioration in cosmetic results. These data offer compelling support that even single fraction radiotherapy administered with MRgRT can offer favorable tumor control and cosmesis without excessive toxicity. Furthermore, a preoperative regimen with ablative doses was found to yield a complete response and a significant proportion of patients, which potentially opens the door to delaying or omitting surgery entirely for interested patients or patients who are not operative candidates.

### 3.5. Lung

Due to significant motion associated with the respiratory cycle and proximity to several critical structures including the heart, normal lung, great vessels, trachea, and esophagus, lung tumors offer a prime example of how MRgRT providing adaptive replanning and motion management techniques can improve clinical care, particularly for central and ultracentral tumors. Motion management techniques are critical to enabling the safe and precise delivery of SBRT to the lung, an organ with continuous intrafraction motion. It is known that achieving a BED ≥ 100 Gy is associated with superior outcomes when treating lung tumors [37]. However, a central location can preclude reaching that objective using traditional techniques. Notably, a dose of 56 Gy in 8 fractions was found to have excessive toxicity [38], so achieving such a dose is challenging. MRgRT provides the opportunity to transition from ITV-based planning to more active monitoring techniques that allow for reduced target volumes, thereby providing potential for improved OAR sparing and minimizing the risk of radiation-induced lung injury.

MRgRT has been shown to be useful in the treatment of thoracic oligometastases, which have the potential to have narrow therapeutic indices due to proximity to central vascular structures and airways. In a phase I trial, Henke et al. report on five patients who had ultracentral malignancies and underwent SMART. The target volume was defined as a 5 mm expansion of the GTV [27]. The prescription dose was 50 Gy in 5 fractions. Daily adaptive planning was available and was used on a total of 10 of 25 total delivered fractions. Of these adaptations, 70% were for reversal of an OAR violation while 30% were for improved PTV coverage. Local control was 100% at 6 months and no acute grade ≥ 3 toxicity was observed. Another study also looked at 50 patients with high-risk lung tumors (29 primary and 21 metastatic) [28]. A risk-adapted dose regimen from 48 Gy in 4 fractions to 54 Gy in 3 fractions was used, with on-table adaptation. Among patients with lung metastases, 1-year LC, DFS, and OS rates were 95.5%, 95.2%, and 57.1%, respectively. The incidence of grade ≥ 2 and grade ≥ 3 toxicity was 30% and 8%, respectively. In a similar series by Regnery et al., 23 primary and secondary lung tumors, of which 11 were ultracentral, were treated with MRgRT [29]. In these plans, <10% of fractions in ultracentral tumors violated OAR constraints, and it was primarily cases where the PTV touched an OAR that violations were seen, representing a key demographic that may benefit from on-table adaptation. Subsequent studies of SMART for peripheral lesions showed much less benefit, suggesting that only select scenarios may require this technology, such as multiple targets or comorbidities [30].

MRgRT has also shown promise for delivery of lung SBRT in a single fraction, which is associated with a high BED and greater patient convenience but must be administered carefully due to no longer having safeguards against excessive toxicity and geographic misses associated with fractionation. In another study by Finazzi et al., single fraction lung SBRT using SMART was evaluated in 17 patients [31]. The GTV was treated with a 5 mm PTV margin and a 3 mm gating window. The treatment time was a median of 120 min and treatment was delivered in two 17 Gy parts with a breath-hold 3DMR scan obtained before each half and the option to adapt. Seven patients were ultimately not suitable for single fraction SBRT due to suboptimal tracking from nearby blood vessels or small tumors < 1 cm. Thus, while single fraction SBRT with MRgRT is clearly feasible, improvements in workflow and imaging capabilities are clearly still needed.

### 3.6. Oligometastases

MRgRT also has a role in the management of oligometastatic disease, where SBRT has been shown to improve PFS and OS [39,40]. Notably, on SABR-COMET, the seminal trial on SBRT for oligometastatic disease, grade ≥ 2 toxicity was observed in 29 patients, with grade 5 toxicity occurring in 4.5% of patients [39]. Therefore, treatment of oligometastatic disease could stand to benefit from reduced PTV margins, which can be afforded by MRgRT. Liver and lung metastases are included in the discussion above but are the focus of several other studies [20,22,23,24].

The treatment of oligometastatic lymph nodes is of interest to a study by Cuccia et al. [32]. In this study, 23 patients with 30 abdomino-pelvic lymph node oligometastases were treated with high-field MRgRT to a total dose of 35 Gy in 5 fractions. Daily adaptive MRgRT was associated with a significant improvement in intestinal loop sparing, reduction in bowel Dmax, and minor improvement in target coverage.

In one of the largest series of MRgRT for abdomino-pelvic metastases, Yoon et al. reported on 106 patients with 121 tumors treated with SMART [33]. A median dose of 40 Gy in 5 fractions was used. A total of 510 fractions were delivered, with 71 (13.9%) being adapted. As a result, only 0.9 and 0% of patients experienced acute grade three and four toxicities, respectively, while 5.2 and 2.1% of patients experienced late grade three and four toxicities, respectively. To highlight the value of dose escalation facilitated with MRgRT, 2-year LC was 96% for lesions that were treated with BED ≥ 100 versus 69% for BED < 100 (*p* = 0.02). PFS was significantly improved in patients with locally controlled tumors (2-year PFS 21 vs. 8%, *p* = 0.03)

Of course, MRgRT is not limited to metastases in these locations and also has shown utility in a number of other sites including breast, adrenal glands, brain, spine, and pediatric patients [41]. Any tumor with close proximity to critical organs, with significant motion or without clear borders on CT, stands to benefit from MRgRT by ensuring that ablative doses required to improve PFS and OS can be delivered.

### 3.7. Cardiac Ablation

An emerging field where MRgRT may have a role in the management of nonmalignant disease is cardiac ablation, where radiation has recently been shown to be an option for the noninvasive treatment of ventricular tachycardia (VT) [42]. The rationale for MRgRT comes from the need to accurately delineate particular areas of interest within the heart, which is made far more feasible with the soft tissue contrast afforded by MRI. Furthermore, the ability to monitor intra-fraction cardiac motion is highly beneficial in ensuring accurate treatment of the target volume. Early feasibility studies suggest cardiac motion can reduce target coverage by 0.1 to 1.3 Gy, which can be effectively mitigated by gating using MRgRT [43].

The literature on the use of MRgRT for cardiac ablation is currently limited to a case report by Mayinger et al. [34]. In this report, a patient with recurrent therapy refractory sustained VT with repetitive implantable cardioverter defibrillator (ICD) shocks was treated with a single fraction of 25 Gy using gating. The total treatment time was 46 min with a beam-on time of 24 min due to treatment gating. Although there was initially a flare of VS, following a high dose of dexamethasone the patient was free of VT after 3 months of follow-up. Clearly more data is needed on how MRgRT may be useful for cardiac ablation, and considerations of how it would interact in patients with implanted pacemakers are also relevant. Nevertheless, MRgRT technology certainly appears to lend itself to the high technologic needs of cardiac ablation with external beam radiotherapy.

## 4. Future Directions

### 4.1. Imaging Improvements

A logical step to improve the benefit of MRgRT is to maximize the information made available within the MRI images. Although there is certainly more soft tissue information available and an MRI compared to a CT, MRI is still susceptible to artifacts, particularly during motion. Therefore, finding ways either in acquisition or post-processing to reduce artifacts could overall improve treatment plans. Additionally, different MR imaging sequences have different utilities depending on the clinical scenario. For example, T2 weighted MRI is most useful for delineating cervical cancer GTV, while T1 contrast-enhanced imaging is optimal for many CNS diseases/metastases. Optimizing imaging sequences to provide the most information for target delineation as well as potential treatment response could further enhance the efficacy of MRgRT.

Specific to response assessment, functional imaging is also of interest. Functional imaging is already being incorporated into radiation oncology workflows in the form of biologically guided radiation therapy using integrated positron emission tomography (PET)-based linear accelerators [44]. MRI functional imaging has potential as well. Diffusion-weighted MRI (DW-MRI) can potentially be correlated with necrosis or apoptosis, and therefore treatment response [45]. Thus, it might be used to select patients based on early response for treatment escalation or de-escalation where appropriate. In a proof of concept, one study utilized DWI imaging to assess response in patients undergoing neoadjuvant therapy for rectal cancer [46]. Other imaging sequences are intended to measure the tumor microenvironment such as hypoxia, which can also help tailor radiotherapy by altering the management of tumors that may be more radioresistant [47]. Lastly, radiomics and quantitative imaging might also be used to monitor treatment response and intervene if needed [48,49].

### 4.2. Adaptive Workflows

MRgRT, and adaptive planning in particular, is associated with additional work compared to traditional non-adaptive CT-based treatment. Future work will need to determine how best to optimize adaptive workflows in order to make routine clinical adoption feasible. The current workflow involves several steps including re-segmentation and, if needed, re-optimization. Given that both adjacent OARs and targets can change on a daily basis, both must be addressed in the adaptive workflow. Existing MRgRT planning systems allow for multiple individuals to work on a single patient at once as part of a parallel adaptive workflow. Therefore, the monitoring of target volumes, OARs, and performance of quality assurance checks can all be performed simultaneously. However, these are still rate-limiting steps and further optimization could lead to new opportunities to speed up the workflow. Options for contouring could include artificial intelligence-based auto-segmentation on a daily basis to provide an improved starting point for contours relative to transferring contours over from the initial simulation. An alternative is to have an off-site team of experts whose job it is to contour relevant structures. Either approach can take the burden off the treating physicians, who may have limited time during the clinic to replan cases on a daily basis.

Another potential for improvement would be to develop decision support tools for deciding when to adapt. In many cases, anatomic changes may not be significant enough to require an adaptation, but the threshold for that decision can be difficult to ascertain without quantitative measures. Automated decision support tools could analyze the initial treatment plan relative to daily MRI setup imaging to provide guidance on which fractions should be adapted and which fractions can be delivered as is.

## 5. Conclusions

MRgRT represents an exciting and versatile emerging technology that has the potential to significantly improve how radiation can manage both primary and metastatic tumors, including the indications detailed in this review as well as head and neck, colorectal, brain, and gynecological cancers. As the field of radiotherapy continues to trend towards more hypofractionated high-dose regimens, the importance of MRgRT in improving the therapeutic index to limit major toxicity is significant. Though there is currently significant clinical evidence for the incorporation of MRgRT, future work will seek to demonstrate its superiority over current standard of care options as well as to streamline associated workflows to make MRgRT and adaptive planning a part of routine clinical practice.

## Figures and Tables

**Figure 1 cancers-15-02916-f001:**
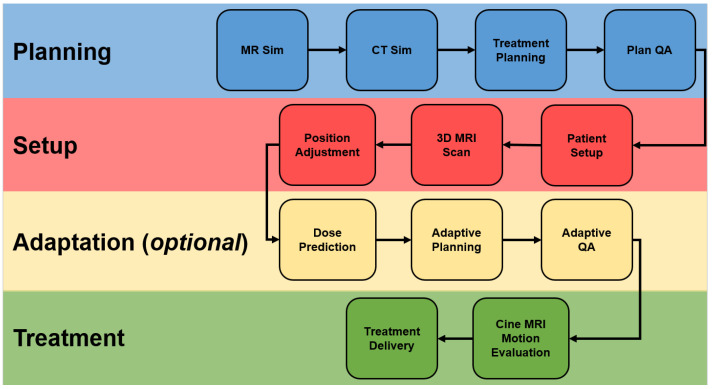
MRgRT. Abbreviations: MR, magnetic resonance; CT, computed tomography; QA, quality assurance; MRI, magnetic resonance imaging.

**Table 1 cancers-15-02916-t001:** Comparison of types of MRgRT.

Characteristic	Low-Field	High-Field
Magnet Strength	0.2–1.0 Tesla	1.5–3.0 Tesla
Cost	Less	More
Space Requirement	Less	More
Compatibility with Metal Implants	More	Less
Commercially Available Models	Viewray MRIdian MagnetTx Aurora RT	Elekta Unity
Signal-to-Noise Ratio	Reduced	Increased
Resolution	Decreased	Increased

**Table 2 cancers-15-02916-t002:** Rationale for MRgRT by disease site.

Disease Site	Enhanced Soft Tissue Visualization	Motion Management	Inter-Fraction Adaptive Re-Planning	Margin Reduction	Facilitate Dose Escalation	Relevant Studies
Prostate	✓			✓		Kishan et al. [15] Bruynzeel et al. [16] Ma et al. [17]
Pancreas	✓	✓	✓		✓	Rudra et al. [18] Parikh et al. [19]
Liver	✓	✓	✓		✓	Rosenberg et al. [20] Luterstein et al. [21] van Dams et al. [22] Ugurluer et al. [23] Henke et al. [24]
Breast	✓	✓		✓		Kennedy et al. [25] Vasmel et al. [26]
Lung		✓	✓			Henke et al. [27] Finazzi et al. [28] Regnery et al. [29] Finazzi et al. [30] Finazzi et al. [31]
Oligometastases	✓	✓	✓	✓	✓	Cuccia et al. [32] Yoon et al. [33]
Cardiac Ablation	✓	✓				Mayinger et al. [34]

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
