# Peer review of "Clinical Applications of Magnetic Resonance-Guided Radiotherapy: A Narrative Review"

_cancers, 2023, doi:10.3390/cancers15112916_

Round 1

Reviewer 1 Report

This review described the clinical indications of MRgRT, which was an important advanced technique in recent years. The manuscript was well written in general. My main comment is that the author just focuses on 7 clinical indications, but it seems not enough. Other main tumor sites such as head and neck cancers, rectum cancers, brain tumors, gynecological cancers should be discussed in this work.       

Author Response

  1. This review described the clinical indications of MRgRT, which was an important advanced technique in recent years. The manuscript was well written in general. My main comment is that the author just focuses on 7 clinical indications, but it seems not enough. Other main tumor sites such as head and neck cancers, rectum cancers, brain tumors, gynecological cancers should be discussed in this work.
    • We would like to thank reviewer 1 for taking the time to review out manuscript. We agree that there are certainly other indications that may be treated with MRgRT. However, this is not intended to be a comprehensive review and rather is intended to focus on the principles of how MRgRT might improve radiation workflows, several of which would also apply to head and neck cancers, rectum cancers, brain tumors, gynecological cancers. To include all disease sites for which there are currently data for MRgRT would require the review to be potentially unwieldy (it is already over 5000 words) and redundant. We have instead decided to focus on the indications included in the original manuscript. We do think this point is valuable though. Therefore we have stated in section 3 that the included indications are not indented to be comprehensive. We have included relevant references if the reader would like to review those indications further. We also highlight these additional disease sites in the conclusion.

Reviewer 2 Report

Dear Editor,

I am pleased to review this fascinating manuscript, which offers a valuable examination of "Clinical Applications of Magnetic Resonance Guided Radiotherapy." While the manuscript is engaging, there are several points that need to be addressed to further enhance its readability and clarity.

1.      I kindly request the addition of a figure to illustrate the clinical workflow of MRI-guided radiotherapy, which will help readers better understand the process.

2.      The use of low-tesla MRI in a clinical setting may result in lower resolution, potentially impacting the accuracy of contouring. Please provide an explanation as to why low-tesla MRI is considered feasible, given the availability of high-tesla MRI options on the market.

3.      MRI-LINAC has been employed in cervical cancer brachytherapy. I recommend adding a section on cervical cancer, referencing the following source: Belardo, Jacob Alexander, et al. "Integrating a Novel GYN Brachytherapy Workflow Using In-Room CBCT Images with MR-LINAC Scans." Brachytherapy 21.6 (2022): S52-S53.

4.      The Lorentz force issue is a notable concern in the context of MR-LINAC. Please include a section in the introduction that explains how this issue can be resolved, further strengthening the manuscript's comprehensiveness.

Author Response

  1. I kindly request the addition of a figure to illustrate the clinical workflow of MRI-guided radiotherapy, which will help readers better understand the process.
    • We would like to thank reviewer 1 for taking the time to review out manuscript. We have added a figure detailing the workflow of MRI-guided radiotherapy.
  2. The use of low-tesla MRI in a clinical setting may result in lower resolution, potentially impacting the accuracy of contouring. Please provide an explanation as to why low-tesla MRI is considered feasible, given the availability of high-tesla MRI options on the market.
    • The pros and cons of low versus high-field MRI are detailed in Table 1. You are right that aside from compatibility of metal implants, greater cost, and greater space requirement, high field would be preferred. However, whether high field MRgRT is accessible, particularly to institutions that already have a low-field MRI-linac, is nuanced and there is certainly still value to low-field MR linacs. We have added to the text however that if available, high-field is in most circumstances preferable.
  3. MRI-LINAC has been employed in cervical cancer brachytherapy. I recommend adding a section on cervical cancer, referencing the following source: Belardo, Jacob Alexander, et al. "Integrating a Novel GYN Brachytherapy Workflow Using In-Room CBCT Images with MR-LINAC Scans." Brachytherapy 21.6 (2022): S52-S53.
    • Thank you for this comment. We have addressed this comment mostly with our response to Reviewer 1, which we have copied to the next bullet point for convenience (which we will follow with additional comments).
    • We would like to thank reviewer 1 for taking the time to review out manuscript. We agree that there are certainly other indications that may be treated with MRgRT. However, this is not intended to be a comprehensive review and rather is intended to focus on the principles of how MRgRT might improve radiation workflows, several of which would also apply to head and neck cancers, rectum cancers, brain tumors, gynecological cancers. To include all disease sites for which there are currently data for MRgRT would require the review to be potentially unwieldy (it is already over 5000 words) and redundant. We have instead decided to focus on the indications included in the original manuscript. We do think this point is valuable though. Therefore we have stated in section 3 that the included indications are not indented to be comprehensive. We have included relevant references if the reader would like to review those indications further. We also highlight these additional disease sites in the conclusion.
    • Our additions in response to reviewer 2 include two different references on MRgRT for cervical cancer. The abstract by Belardo et al. does indeed use the MRI imaging capabilities of the MRI Linac. However, this is essentially used in place of a MRI scanner for treatment planning and does not use the radiation delivery functionality of the MRI linac. We therefore feel this specific citation is beyond the scope of this review, as it does not involve delivery of MR-guider external beam RT.
  4. The Lorentz force issue is a notable concern in the context of MR-LINAC. Please include a section in the introduction that explains how this issue can be resolved, further strengthening the manuscript's comprehensiveness.
    • The Lorenz force is indeed an important concern for MR-linacs. We have added text and a reference detailing how MR-linac planning systems are specifically designed with this concern in mind. A detailed explanation of the Lorentz force is beyond the scope of this review, which is primarily focused on clinical applications.

Round 2

Reviewer 1 Report

my comments were well addressed.

Reviewer 2 Report

Dear editor, 

I am glad to review this manuscript again. As indicated in the first round review report, the indication of MR guided RT in this review is too few. I know the authors argue that adding other indications will make the review too lengthy. However, I still think the authors should try to add more indications in the articles, which could make the reader more clear about the indications of MR guided RT.